# Acceptability and feasibility of an acceptance and commitment therapy-based guided self-help intervention for weight loss maintenance in adults who have previously completed a behavioural weight loss programme: the SWiM feasibility study protocol

Amy L Ahern [ID],[1] Rebecca Richards [ID],[1] Rebecca A Jones,[1] Fiona Whittle,[1] Julia Mueller [ID],[1] Jenny Woolston,[1] Stephen J Sharp,[1] Carly A Hughes,[2,3] Andrew J Hill,[4] Robbie Duschinsky [ID],[5] Emma Ruth Lawlor [ID],[1] Stephen Morris [ID],[5] Francesco Fusco,[5] Alan Brennan,[6] Jennifer Bostock,[7] Simon J Griffin[1]

For numbered affiliations see end of article.

**Correspondence to**
Dr Amy L Ahern;
ala34@cam.ac.uk

## ABSTRACT

**Introduction** The cost-effectiveness and long-term health impact of behavioural weight management programmes depends on post-treatment weight-loss maintenance. Growing evidence suggests that interventions using acceptance and commitment therapy (ACT) could improve long-term weight management. We developed an ACT-based, guided self-help intervention to support adults who have recently completed a behavioural weight loss programme. This study will assess the feasibility and acceptability of this type of intervention and findings will inform the development of a full-scale trial.

**Methods and analysis** This is a pragmatic, randomised, single-blind, parallel group, two-arm, feasibility study with an embedded process evaluation. We will recruit and randomise 60 adults who have recently completed a behavioural weight loss programme to the ACT-based intervention or standard care, using a computer-generated sequence with 2:1 allocation stratified by diabetes status and sex. Baseline and 6-month measurements will be completed using online questionnaires. Qualitative interviews will be conducted with a subsample of participants and coaches about their experiences at 3 (mid-intervention) and 6 (postintervention) months. Feasibility and acceptability of the intervention, and a full-scale trial will be assessed using a number of outcomes, including adherence to, and engagement with the intervention, recruitment and retention rates, proportion of missing data for each outcome measure, participants' experiences of the intervention and study, and coaches' experiences of delivering intervention support. Quantitative and qualitative findings will be integrated and summarised to contribute to the

### Strengths and limitations of this study

► This is the first study to assess the feasibility and acceptability of a web-based, guided self-help acceptance and commitment therapy-based intervention to support weight loss maintenance in adults who have recently completed a behavioural weight loss programme in the UK.

► This study uses mixed methods and will draw on both quantitative and qualitative data to assess the feasibility and acceptability of the intervention.

► This study includes an embedded process evaluation to identify what worked, what did not and why, including mid-intervention (3 months from baseline) and postintervention (6 months from baseline) qualitative interviews with both participants and coaches to better understand their experiences throughout the intervention period.

► The study and intervention are conducted solely online and will provide insight into conducting the intervention and future trial remotely.

► This study is limited by self-reported weight as a primary outcome measure.

interpretation of the main feasibility evaluation findings. Value of information methods will be used to estimate the decision uncertainty associated with the intervention's cost-effectiveness and determine the value of a definitive trial.

**Ethics and dissemination** Ethical approval was received from Cambridge South Research Ethics Committee on 15/03/2021 (21/EE/0024). This protocol (V.2) was approved on 19 April 2021. Findings will be

## INTRODUCTION

The cost-effectiveness and long-term health impact of behavioural weight management programmes depends on post-treatment weight loss maintenance.[1] Systematic reviews show that most weight is regained within 3–5 years, even after specialist-led behavioural programmes.[2 3] Extended use of traditional behavioural strategies (eg, self-monitoring, problem solving) can improve weight loss maintenance to some extent,[4] but new approaches are needed to maximise the benefits of behavioural weight management programmes.

There is growing evidence to suggest that interventions that incorporate strategies from acceptance and commitment therapy (ACT) may be effective for long-term weight control and can improve some psychological determinants of weight loss maintenance.[5 6] However, to date, most studies have been conducted in-person, in a US setting and the cost-effectiveness of this type of intervention has not been evaluated. In addition, ACT-based interventions are usually psychologist led and the cost and scarcity of psychologists specialising in obesity could limit their use in countries with a national healthcare system. There is currently insufficient evidence on the potential scalability of an ACT-based intervention that is delivered remotely to support weight loss maintenance in the UK, including the importance of facilitator expertise and cost-effectiveness. To address this, we developed the Supporting Weight Management (SWiM) programme as a web-based, guided self-help intervention that uses ACT-based treatment and specifically focuses on supporting post-treatment weight loss maintenance. SWiM uses digital technology and non-specialists (referred to here as 'SWiM Coaches') to minimise resources needed to deliver an ACT-based intervention at scale. SWiM is intended to be used following completion of a standard behavioural weight management programme (lasting at least 12 weeks), and seeks to reinforce what helped people to lose weight, build on what worked for them, and teach new ACT-based skills and strategies to support them in the longer term.

The study described in this protocol is designed to assess the feasibility and acceptability of the SWiM intervention, and inform the development of a full-scale trial. The first section of this protocol describes the feasibility evaluation, which focuses on the feasibility of conducting a full-scale trial comparing SWiM to standard care. The subsequent section describes the embedded process evaluation, guided by the MRC framework for process evaluations of complex interventions in healthcare.[7] The process evaluation focuses on assessing the implementation of the intervention, identifying the contextual factors that may be associated with variations in implementation and outcomes, as well as exploring the hypothesised causal mechanisms of the intervention. These findings will provide insight into what worked, what did not and why, and enable us to identify and implement further refinements needed to improve the intervention for a future trial.

## Aims and objectives
### Aim

The aims of this study are to: (1) inform the development of a future trial which evaluates effectiveness and cost-effectiveness by minimising uncertainties about trial parameters, and (2) evaluate the feasibility and acceptability of the SWiM intervention.

### Objectives
*Feasibility and acceptability of a future trial*

1. To assess the feasibility of a full-scale trial:
   – To examine feasibility and acceptability of recruiting participants directly from existing weight management programmes and identify effective methods that minimise participation bias.
   – To demonstrate feasibility and acceptability of randomisation procedures.
   – To estimate recruitment and retention rates for a full scale (cost-)effectiveness randomised controlled trial.
   – To test systems for data collection and outcome assessment.
2. To estimate parameters required to inform the sample size and value of a full-scale trial:
   – To describe changes in outcome variables and estimate variance to inform the design of a full-scale trial.
   – To use value of information (VOI) methods to estimate the decision uncertainty associated with the intervention's cost-effectiveness and determine the value of a definitive trial, based on ability to reduce decision uncertainty.

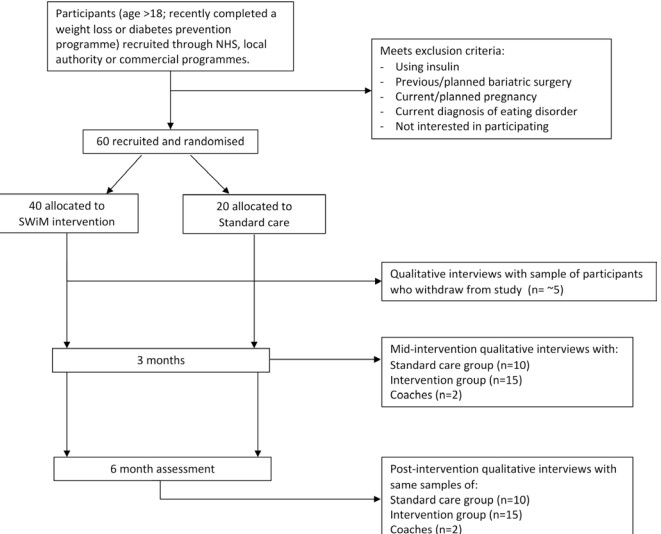

**Figure 1** Trial flow chart. SWiM, supporting weight management; NHS, National Health Service

*Process evaluation*

3. To assess the acceptability and feasibility of delivering the SWiM intervention using the web-based platform and trained non-specialists (SWiM Coaches) in a community setting:

    – To assess the implementation of the intervention in terms of reach (whether the intended audience came into contact with the intervention), fidelity (the quality of what was delivered) and dose (the quantity of intervention delivered).

    – To identify how contextual factors influence variations in implementation and outcomes, and identify barriers and facilitators to delivery.

    – To explore participants' experiences of the intervention, including their views on acceptability, their patterns of use, facilitators and barriers to use, and the extent to which it meets their needs.

    – To explore the SWiM Coaches' experiences of delivering the support, including their views on acceptability, their training and support needs, and the facilitators and barriers to delivering support.

    – To explore how participant individual characteristics influence acceptability and engagement.

    – To qualitatively explore the hypothesised causal mechanisms of SWiM and any unexpected mechanisms.

    – To identify ways to further develop and optimise the content and delivery of the SWiM intervention for a future trial.

## METHODS AND ANALYSIS

This is a pragmatic, randomised, single-blind, parallel group, two-arm, feasibility study with an embedded process evaluation (figure 1). Participants will be randomised to either the ACT-based weight maintenance intervention (SWiM) or to standard care, using a computer-generated sequence with 2:1 allocation stratified by diabetes status and sex. This allocation will allow us to observe a reasonable number in the standard care group to gain information about their experiences, estimate retention in this group and assess willingness to be randomised. Participants will complete baseline and 6-month (from baseline) measurement sessions using online questionnaires. A subsample of participants from the standard care and intervention groups, and all coaches, will be invited to participate in qualitative interviews at both 3 months (mid-intervention) and 6 months from baseline (post-intervention). In addition, a subsample of participants who enrol but withdraw from the intervention group will also be interviewed. Qualitative interviews will collect data that will be used to contribute to both the feasibility evaluation and the process evaluation.

## Participants

We will recruit adults (N=60) who have recently completed a behavioural weight loss programme lasting at least 3 months. Eligible participants will be recruited through NHS, local authority and commercial weight management services and diabetes prevention programmes in the UK, opportunistically, by referral and by mail out.

## Inclusion criteria

1. Age ≥18 years.
2. Completed a behavioural weight loss programme in the last 3 months.
3. Capable of giving informed consent.
4. Have a good understanding of the English language (for the feasibility study, materials are not tailored to support non-English language speakers).
5. Willing to be randomised.
6. Willing to complete study measurements.
7. Able to access the web-based platform from home.
8. Own a set of scales that they can weigh themselves with during the study.

There is no requirement for verification of participation in a behavioural weight loss programme in the last 3 months in order to avoid unnecessary additional burden on participants or referring programmes, and to support efficient referral and wide uptake. There are no criteria for having lost weight during a behavioural weight loss programme as the SWiM programme can be used to support and build on participants' learnings from their behavioural weight loss programme and participants can develop further skills and strategies from ACT that could help them to lose weight or prevent further weight gain.

## Exclusion criteria

Based on expert stakeholder advice that the specific support needs of the following groups are beyond the remit of this intervention, adults who meet any of the following criteria will not be eligible for inclusion:

1. Using insulin.
2. Previous or planned bariatric surgery.
3. Current or planned pregnancy.
4. Current diagnosis of eating disorder.

There are no inclusion or exclusion criteria for body mass index (BMI) in order to be as inclusive as possible. Participants typically require a BMI of 25 kg/m$^2$ or higher (or 23 kg/m$^2$ depending on ethnicity) to be eligible for behavioural weight loss programmes. Therefore we do not expect that participants will lose enough weight to reduce their BMI to 18.5 kg/m$^2$ or less (ie, 'underweight') and participate in the SWiM programme. Any potential participants who are underweight would be identified at screening and further assessment would be conducted to determine suitability.

We expect that some participants may continue to receive weight management support while participating in the SWiM programme (eg, attending groups, using an app to monitor food intake) and we do not wish to exclude these participants. We will collect information on whether participants continue to receive any support from their original behavioural weight loss programme or additional interventions. If participants report that they are receiving a more intense intervention, such as very

low calorie diets, or a medical intervention, such as GLP-1 analogues, during the feasibility study, we will judge their eligibility to participate and/or continue on a case-by-case basis. This feasibility study will inform whether further inclusion and exclusion criteria should be implemented prior to a full trial.

### Recruitment

Participants will be recruited through NHS, local authority and commercial weight management services and diabetes prevention programmes. Eligible people will be given or mailed a study information leaflet by their programme leader within their weight or type 2 diabetes management programme. Those who are willing to participate will either give permission for the programme leader to pass on their details, or they will be asked to contact the study team directly by telephone or email for more information and to ask any questions. If participants are willing to take part, they will be sent a secure web form which they will use to provide participant information, confirm eligibility and provide informed e-consent (online supplemental file 1). Once informed consent is received, baseline data will be collected.

### Randomisation

Participants will be allocated to one of the two intervention arms in a 2:1 allocation using block randomisation stratified by type 2 diabetes status and sex (male, female). The randomisation sequence will be computer generated by the trial statistician and programmed by the data manager. The sequence will be unknown to all other personnel, including study coordinators, outcome assessors and investigators. Once eligibility is confirmed and the online baseline assessment complete, the randomised allocation will be revealed to the participant by phone or email by the study coordinator. Intervention materials will then be provided depending on the group allocation. For logistical reasons, the study coordinator and the data manager will not be blinded to the allocation group. The trial statistician and the Investigators will be blinded to intervention allocation until the database is locked and the primary analysis complete.

### Planned intervention and standard care
#### SWiM intervention
SWiM is a web-based, guided self-help intervention that uses ACT-based treatment to support adults following the completion of a behavioural weight loss programme. It aims to help them to reflect on what has worked (and not worked) in the past, build on what works for them, and learn new ACT-based skills and strategies to overcome challenges that typically derail weight loss maintenance. The intervention includes access to an online web platform with 14 sessions ('SWiM sessions') consisting of psychoeducational content, reflective exercises and behavioural experiments. SWiM consists of weekly sessions for the first 13 consecutive weeks, followed by a 4-week break for

participants to reflect and practice their new skills and strategies, then a final session at week 18 (table 1). Each session is expected to take around 30–60 min to complete. Participants are encouraged to weigh themselves weekly and to record their weight at the start of each session. Between sessions, participants are asked to complete more reflective exercises and behavioural experiments (called 'SWiM Practice'). Full details of intervention content have been published elsewhere.[8]

To guide them through the intervention, participants receive four scheduled telephone support calls from a trained, non-specialist 'SWiM coach' over the course of the intervention (specialists being defined as professionals with specialist qualifications or registration in weight management, for example, dieticians). Coaches are trained by a member of the research team, who is a practitioner psychologist (RR), and the training was developed by the research team specifically for this study. As part of the training, coaches are asked to: (1) read through a training manual, which includes semistructured scripts to guide telephone calls and (2) attend 3 hours of training with the practitioner psychologist either in person or via video call. The training manual includes an outline of intervention content, underpinning theories of the SWiM programme (ACT and motivational interviewing), a practical guide to conducting telephone support, the procedures for participant withdrawal, information on how to avoid stigmatising language and information for signposting participants to mental health support. The practitioner psychologist training takes coaches through the training manual in detail, provides opportunities for questions and includes role-play of each telephone call script with feedback. Following completion of this training, coaches have the opportunity for one-to-one or group follow-up sessions for further practice or clarifications. Coaches are also asked to complete each session on the SWiM website from the perspective of the participant.

Calls are scheduled following completion of sessions 1, 3, 8 and 14. There will also be three additional, optional calls that can be used at any time should the participant want further support from the coach. The role of the coach is to help the participant take ownership of their weight management. Automated email reminders are sent to participants to remind them to complete sessions and calls.

#### Web platform
On the SWiM web platform, intervention content is divided into SWiM sessions, which are each divided into activities. Progress through the sessions is presented as a 'journey' using a map-like graphic. Star icons light up when activities and sessions are completed. As participants complete core skills and exercises, these are stored in a 'SWiM Aids' tab, where they can be accessed without revisiting specific sessions. The web platform allows participants to revisit past sessions and skills, and

| Table 1 | SWiM intervention outline and content |
|---|---|
| **Session** | **Content** |
| Welcome to SWiM | Let's take a look around SWiM! |
| | Meet the SWiM team |
| | Your Commitment to SWiM |
| Session 1: planning and tracking | 1.0 What is SWiM? |
| | 1.1 Your weight maintenance plan |
| | 1.2 Tracking your progress |
| | 1.3 SMART goals and plans |
| | 1.4 SWiM practice: goal setting |
| Session 2: control and acceptance | 2.0 Checking in |
| | 2.1 Control and acceptance |
| | 2.2 What matters to you? |
| | 2.3 SWiM practice: values, goals and actions |
| Session 3: being willing | 3.0 Checking in |
| | 3.1 Values and goals |
| | 3.2 Being willing |
| | 3.3 SWiM practice: 'Even If…' Thoughts |
| Session 4: overcoming obstacles | 4.0 Checking in |
| | 4.1 Identifying your obstacles |
| | 4.2 Planning for obstacles |
| | 4.3 SWiM practice: being BOLD |
| Session 5: being active and willing | 5.0 Checking in |
| | 5.1 Physical activity recommendations |
| | 5.2 Obstacles to being active |
| | 5.3 Applying willingness to physical activity |
| | 5.4 SWiM practice: your physical activity plan |
| Session 6: emotional eating | 6.0 Checking in |
| | 6.1 What is emotional eating? |
| | 6.2 Breaking the cycle |
| | 6.3 SWiM practice: emotional responses diary |
| Session 7: stress management | 7.0 Checking in |
| | 7.1 Stress and weight gain |
| | 7.2 Control what you can, accept what you can't |
| | 7.3 Defusion: unplugging the sink |
| | 7.4 Mindful breathing |
| | 7.5 SWiM practice: practising defusion |
| Session 8: forming helpful habits | 8.0 Checking in |
| | 8.1 Recap of sessions 1 to 7 |
| | 8.1 Forming helpful habits |
| | 8.2 SWiM practice: forming your new habit |
| Session 9: breaking unhelpful habits | 9.0 Checking in |
| | 9.1 Breaking unhelpful habits |
| | 9.2 Being flexible |
| | 9.3 SWiM practice: breaking your unhelpful habits |
| Session 10: urges and cravings | 10.0 Checking in |
| | 10.1 We all have urges and cravings |
| | 10.2 A recap of defusion |
| | 10.3 Urge surfing |
| | 10.4 SWiM practice: learning to surf |

| Table 1 | Continued |
|---|---|
| **Session** | **Content** |
| Session 11: the power of sleep | 11.0 Checking in |
| | 11.1 The power of sleep |
| | 11.2 Sleep and weight management |
| | 11.3 How to get a good night's sleep |
| | 11.4 SWiM practice: forming helpful sleep habits |
| Session 12: friends and family | 12.0 Checking in |
| | 12.1 Friends and family |
| | 12.2 How to get the support you need |
| | 12.3 Breaking unhelpful food rules |
| | 12.4 SWiM practice: rule breaking and being assertive |
| Session 13: weight stigma and body image | 13.0 Checking in |
| | 13.1 wt stigma |
| | 13.2 How to deal with weight stigma |
| | 13.3 Body image |
| | 13.4 Self-acceptance |
| | 13.5 Physical activity and body image |
| | 13.6 SWiM practice: practicing self-acceptance |
| Session 14: lapses and maintaining motivation | 14.0 Checking in |
| | 14.1 Lapse vs relapse |
| | 14.2 Strategies to prevent a relapse |
| | 14.3 Reversing small weight gains |
| | 14.4 Maintaining motivation |
| | 14.5 Going forward |

SWiM, supporting weight management.

it is intended that future roll out would include indefinite access to the website. The web platform includes a weight tracker, which generates a line graph that automatically updates as data is inputted by the participant. As part of the first session, participants are asked to enter their prior weight loss, so tha they can see what they have already achieved. The weight tracker automatically sets a weight maintenance target range with a boundary of ±3 kg that participants are encouraged to stick within. This boundary can be adjusted if required as weight changes over time. Each session starts with a reflection on the previous session and SWiM practice, and entry of weight data into the tracker.

Participants will be given the option of receiving a printed booklet of the website exercises in the post if they prefer to write their answers down, however, they will be encouraged to complete the exercises online as well, so that the coach can provide tailored support.

### Standard care intervention

Participants who are randomised to the standard care intervention will be emailed a leaflet about weight loss maintenance which helps them to make a personalised weight maintenance plan (online supplemental file 2).

**Table 2** Feasibility and acceptability outcomes

| Outcomes | Data source |
|---|---|
| No and characteristics of those invited | Data from weight management services |
| No and characteristics of:<br>► Recruited participants (by recruitment method and by study group)<br>► Respondents (by recruitment method and by study group)<br>► Participants who withdraw or have missing data (by study group) | Recorded in study database |
| Recruitment rate (per month, per weight management service and by study group) | Recorded in study database |
| Proportion of missing data (by outcome measure) | Recorded in study database |
| Experience of participants and coaches in terms of participating in, and supporting the delivery of, the study, respectively. | Qualitative telephone/video interviews |
| Views and experiences of participants who withdraw from the study | Qualitative telephone/video interviews |

**Table 4** Schedule of enrolment, interventions and assessments

| | Time point | | |
|---|---|---|---|
| | Enrolment | Baseline (0 months) | Follow-up (6 months) |
| **Enrolment** | | | |
| Telephone eligibility screen | X | | |
| Informed consent | | X | |
| Randomisation | | X | |
| **Interventions** | | | |
| Supporting weight maintenance | | ●————● | |
| Standard care | | X | |
| **Assessments** | | | |
| Height | | X | |
| Weight | | X | X |
| Self-report questionnaires | | X | X |

## Feasibility and acceptability evaluation of a future trial
### Outcomes and measures
*Feasibility and acceptability*
The feasibility of a full-scale trial will be assessed quantitatively and qualitatively using data from multiple sources (table 2).

### Outcomes for assessment of effectiveness
We will calculate baseline to 6-month changes in outcome variables outlined in table 3. In a potential future trial, the primary outcome measure will be change in weight from baseline to 6 months. Other measures of interest include health-related quality of life (HRQOL) and well-being, economic impact (health resource use, out-of-pocket expenses) and psychosocial factors hypothesised

to be determinants of weight loss maintenance that are targeted by the intervention (disinhibition, food cravings, psychological flexibility).

## Measurements and data collection
### Visit schedule
Participants will be asked to complete online assessments at baseline and 6 months. Details of which measures will be taken at each appointment are summarised in table 4. Participants will be given an honorarium for completing online assessments (£10 for baseline and £20 for the 6-month assessment). Honoraria for assessment attendance are not dependent on intervention attendance or completion. Participants who complete qualitative interviews will be given a £10 honorarium per interview.

**Table 3** Outcomes and measures

| Domain | Outcome | Measure | Time point |
|---|---|---|---|
| Clinical outcomes | Height | Self-measured | 0 months |
| | Weight | Self-measured | 0 months; 6 months |
| Quality of life and well-being | Health-related quality of life | EQ-5D-5L[19] | 0 months; 6 months |
| | Capability/well-being | ICECAP-A[20] | 0 months; 6 months |
| Economic evaluation | Health/social care use | Bespoke resource use questionnaire | 0 months; 6 months |
| | Out of pocket costs | Bespoke resource use questionnaire | 0 months; 6 months |
| Psychosocial factors | Disinhibition | Three-Factor Eating Questionnaire[18] | 0 months; 6 months |
| | Psychological Flexibility | Acceptance and Action Questionnaires (Weight related; food related)[21 22] | 0 months; 6 months |
| | Depression<br>Anxiety<br>Stress | Patient Health Questionnaire 8-item[23 24]<br>Generalised Anxiety Disorder 7-item scale[25]<br>Perceived Stress Scale[26 27] | 0 months; 6 months |
| | Habits | Bespoke questionnaire adapted from the Self-Report Habit Index[28] | 0 months; 6 months |

## Anthropometric measures

All outcomes will be assessed via online self-report questionnaires. Participants will be asked to measure and report their height and to weigh themselves on the day that they complete the outcome assessment so that they can report a self-measured weight. Instructions for self-measuring will be provided to reduce measurement bias.

## Self-report questionnaires

Participants will complete a demographics questionnaire at baseline based on Progress-Plus[9] factors (place of residency, race/ethnicity, occupation, gender/sex, religion, education, socioeconomic status, social capital, age, disability, relationship status, caring responsibilities, car ownership and access to the internet). Self-reported behavioural and psychosocial measures will be collected via validated self-report questionnaires, which will be completed online (table 3).

## Referral data

Data on invited numbers and referrals will be gathered from the recruiting weight management programmes, where possible. Individual strategies to gather these data will be agreed with the weight management/referral service at set-up.

## Data analysis

### Statistical analysis

Quantitative analyses will be primarily descriptive. We will describe the number and proportion of participants who are invited to the study, respond, enrol (by recruitment method) and complete outcome assessments. Each of these will be examined for potential biases by gender, age, ethnicity and socioeconomic status. Change from baseline to 6 months for weight and each of the secondary outcomes will be summarised separately within the intervention and standard care groups using mean (SD) or median (IQR). Linear regression will estimate the difference (intervention minus standard care) in mean change in weight from baseline, adjusted for baseline weight and the randomisation stratifiers (diabetes status and sex). Since this is a feasibility study, it is not powered to detect differences between randomised groups or mediation by psychosocial variables. However, it will provide estimates of the SD of weight at baseline and correlation between weight at baseline and weight at 6 months, which will inform the sample size calculation for a full-scale trial.

VOI methods will estimate the decision uncertainty associated with the intervention's cost-effectiveness given the existing evidence base, and will determine the value of a definitive trial, based on ability to reduce decision uncertainty. Using data from the feasibility study, we will undertake a formal, structured elicitation process to elicit expected trial outcomes and uncertainty. We will use the Sheffield Elicitation Framework methods[10] and recruit participant experts on the likely effectiveness of the intervention from within the research team and externally. If the elicitation suggests very small effects and a high cost then we might already be certain of poor cost-effectiveness. If it suggests large effects and a small cost, this may be sufficient to suggest the intervention is highly cost-effective. However, and most likely, if the intervention lies somewhere between these scenarios, we will use expected value of sample information to estimate the value of alternative trial designs in reducing uncertainty in the cost-effectiveness estimates. This will help to select an optimal trial design to inform future cost-effectiveness analyses. We currently plan to examine three options for sample size (small, medium, large) and follow-up (1.5, 3, and 5 years), however, the exact details will be developed in an iterative process to support the trial design.

## Economic analysis

Economic analyses (cost-effectiveness analysis, cost–utility analysis) will be undertaken, comparing the incremental costs and effects of the SWiM programme versus standard care. We will conduct a microcosting to estimate resource use for the set-up, delivery and maintenance of the SWiM programme. Resource use estimates will be derived from the literature and informed by feasibility study data, and will be adjusted based on expert input through an informal panel process.

Healthcare costs will be calculated as the product of resources used and appropriate unit costs adopting National Health Service (NHS) and personal social service (NHS-PSS) and patientperspectives. Cost components will include the SWiM intervention, primary care (eg, General Practitioner (GP) and nurse visits) and secondary care services (eg, outpatient and hospital admissions). Resource use data will be captured by case report forms (CRF) and a Short Health Economics Questionnaire (SHEQ), the latter also including over-the-counter medications, private visits, travelling and productivity loss. The CRF and SHEQ will be completed at baseline (covering the 3 months before the baseline) and 6 months (covering the 6 months from the baseline) by clinical personnel and patients, respectively. The outcomes used in the cost-effectiveness analysis will be the change in weight over time and HRQOL captured by the EuroQol EQ-5D-5L questionnaire, which will be expressed in quality-adjusted life-years.[11] Missing data on costs and outcomes are anticipated, and will be handled using multiple imputation methods.[12]

Imputed mean cost-effectiveness in each arm will be used to obtain an incremental cost-effectiveness ratio (ICER), which will be calculated by dividing the mean cost difference by the mean effectiveness difference. These mean differences will be estimated using regression methods adjusting for HRQOL, costs and any unbalanced variables observed at randomisation. The uncertainty around the incremental cost and incremental effectiveness estimates will be evaluated using non-parametric bootstrapping methods and presented as cost-effectiveness probabilities, estimated as the proportion of the bootstrapped cost and effectiveness pairs with corresponding ICERs below different values of the cost-effectiveness threshold.

The economic analysis will also have a feasibility component. This will address whether any important cost components should be added to the analysis, the sources of unit cost data, the extent of missing data and methods for dealing with this, and the feasibility of all analytical methods. Second, the results of the probabilistic sensitivity analyses will be combined with the epidemiological information on projected numbers to undertake a VOI analysis to evaluate the potential economic value and acceptability of future research.[13]

### Qualitative analysis

Qualitative analysis procedures are described within the process evaluation section below. For the feasibility evaluation, we will draw out any themes relating to the conduct of the study (rather than the intervention) from qualitative interviews with participants and coaches, such as experiences relating to recruitment methods and completion of outcome measures. We will also conduct a small number of interviews with a sample of participants who enrol in the SWiM intervention but withdraw from the study. This will provide further insight into the enrolment process and barriers to engagement with the study.

## PROCESS EVALUATION

The embedded process evaluation within this feasibility study aims to contribute to the interpretation of the feasibility evaluation findings by assessing the feasibility and acceptability of the intervention and identifying what worked, what did not, and why. We will do this by answering the following research questions, guided by the MRC framework for process evaluations of complex interventions in healthcare.[7]

1. What is implemented and how?
2. How might the intervention produce change?
3. How might context affect implementation and outcomes?

### Outcomes and measures

The key components of the MRC framework for process evaluations of complex interventions in healthcare,[7] and corresponding research questions, will be assessed quantitatively and qualitatively using data from multiple sources (table 5).

### Measurements and data collection

Quantitative data will be collected through the postintervention questionnaire, website analytics, SWiM session feedback questionnaires (from the website), coach telephone call reports (completed by coaches) and study coordination records (eg, recruitment data). Qualitative data will be collected through semistructured telephone/video interviews at mid- (3 months from baseline) and postintervention (6 months from baseline), and open-text questions from participant questionnaires, SWiM session feedback questionnaires, coach support forms (completed by participants) and coach telephone call

reports. The timing of data collection is presented in table 6.

### Postintervention questionnaires

The postintervention (6 months from baseline) questionnaire will collect quantitative and qualitative data on perceived ease of use, usefulness and enjoyment of the intervention, and participants' patterns of use, experiences of, and opinions on, the intervention.

### Website analytics

Data on frequency (how often contact is made with the intervention over a specified period of time), amount (total length of each intervention contact), duration (the period of time over which participants are exposed to an intervention), and depth (variety of content used) will be collected.

### SWiM session feedback questionnaires

Following each completed SWiM session, quantitative data will be collected using questionnaires from the website on participants' perceptions of the content, including how easy the session was to understand, how useful it was and how relevant it was to their weight management. Qualitative data will be collected on participants' experiences of the behavioural experiment at the end of each session.

### Coach support forms

Prior to the scheduled telephone calls with the coach after SWiM sessions 1, 3, 8 and 14, qualitative data will be collected from intervention participants on their experiences of the intervention so far, whether they would like any help from their coach, and whether they had any questions about their weight management.

### Coach telephone call reports

Following completion of each telephone call with a participant, quantitative and qualitative data will be collected from coaches on the type, duration and content of the call, their experiences of delivering the call (eg, how easy it was to complete, whether they were able to deliver it as intended) and how they might improve for next time.

### Qualitative interviews

Qualitative interviews will be conducted over the telephone, or via Zoom video software, at mid-intervention (3 months from baseline) and postintervention (6 months from baseline) with a subsample of intervention (n=15) and standard care participants (n=10) and all SWiM coaches (n=2). Interviews will be conducted at both time points for all groups in order to capture changes over the course of the study. Intervention and standard care participants will be purposively sampled using outcome data (broad demographic, range of weight outcomes). Questions will focus on the extent and manner of intervention implementation, challenges experienced, their experiences of the intervention and their remaining needs at the study end. We also aim to interview up to five participants who withdraw from the intervention arm to

**Table 5** Key components of the MRC framework for process evaluations of complex interventions in healthcare and corresponding key questions

| MRC framework components | Key questions | Data source |
|---|---|---|
| 1. **Implementation:** What is implemented, and how? | **Fidelity:** Assess the quality of what was delivered: ▶ What was delivered? ▶ How was it delivered? ▶ Were there any unexpected changes in implementation? | ▶ Website analytics (frequency, amount, depth and duration of engagement) ▶ Coach call reports ▶ Questionnaires (perceived ease of use, usefulness and enjoyment; technical issues; patterns of use; device/s) ▶ Interviews with participants and coaches ▶ Study coordination records |
| | **Dose:** Assess the quantity of intervention delivered. | ▶ Website analytics (frequency, amount, depth and duration of engagement) ▶ Coach call reports (call completion and duration data) ▶ Interviews with participants and coaches |
| | **Adaptations:** Assess whether there were adaptations to make the intervention fit different contexts: ▶ Did these undermine fidelity? | ▶ Interviews with participants and coaches ▶ Coach call reports ▶ Study coordination records |
| | Reach: ▶ Did the intended audience come into contact with the interventions? How? | ▶ Study coordination records ▶ Demographic data ▶ Website analytics ▶ Interviews with participants and coaches |
| 2. **Mechanisms of impact** How does the delivered intervention produce change? | ▶ How did the effects occur? (Explore hypothesised causal pathways from the logic model) ▶ Were there any unexpected mechanisms of action? | ▶ Website analytics (frequency, amount, depth, and duration of engagement) ▶ Interviews with participants and coaches ▶ SWiM Session feedback questionnaires ▶ Coach call reports |
| 3. **Context** How does context affect implementation and outcomes? | ▶ Are there contextual factors that affect (and may be affected by) implementation, intervention mechanisms and outcomes? ▶ Assess the generalisability of potential effectiveness by understanding the role of context | ▶ Analysis of uptake and adherence by Progress Plus criteria ▶ Questionnaires (open-ended questions, eg, things that made it difficult/helped to complete the intervention) ▶ Interviews with participants and coaches ▶ Coach support forms ▶ Coach call reports |

MRC, Medical Research Council; SWiM, supporting weight management .

explore their experiences of the study and intervention. Interviews with coaches will focus on their experiences of delivering the coach support and their training and support needs throughout the study.

## Analysis
### Quantitative analysis
Process evaluation data will be analysed independently of the main outcome data. We will report key descriptive statistics on intervention adherence (eg, number of sessions completed) and engagement (how participants used the intervention) to provide an overview of reach, dose and fidelity:

1. Descriptive statistics for sample characteristics, informed by the PROGRESS Plus framework (eg, age, sex, education level, marital status, weight, height, BMI).
2. Average number of sessions completed by participants.
3. Number, proportion and characteristics of participants completing the first four sessions and scheduled telephone calls with their coach.
4. Number, proportion and characteristics of participants completing the first nine sessions (two-thirds of the total intervention) and scheduled telephone calls with their coach.
5. Number, proportion and characteristics of participants completing all 14 sessions (ie, the total number of sessions) and scheduled telephone calls with their coach.
6. Number and average duration of scheduled coach calls completed.
7. Number and average duration of additional optional coach calls completed.

**Table 6** Data sources and collection for the embedded process evaluation

| | 0 months (baseline) | Throughout the intervention | 3 months from baseline (mid-intervention) | 6 months from baseline (postintervention) |
|---|---|---|---|---|
| Questionnaires | X | | | X |
| Website analytics | | X | | |
| SWiM session feedback questionnaires | | X | | |
| Coach support forms | | X | | |
| Coach call reports | | X | | |
| Interviews | | | X | X |
| Study coordination records | ◆————————————————————◆ | | | |

SWiM, supporting weight management.

## Qualitative analysis

All qualitative data will be analysed using reflexive thematic analysis.[14] A dual coding approach will used: initial inductive open coding will be carried out to generate themes, followed by a second deductive coding round based on the MRC framework for process evaluations[7] (table 5) and the results of planned quantitative analyses. Both phases of coding will be conducted on each qualitative dataset in succession (eg, interviews, followed by open-text responses of questionnaires). These two stages of analysis will allow the qualitative findings to specifically answer the research questions of the process evaluation, and speak effectively to the questions arising from the quantitative findings, while retaining an integral attention to issues that might otherwise be lost in the qualitative data, or be silent in the quantitative data. In addition, the inductive coding stage ensures that data that does not specifically relate to the MRC framework for process evaluations is not lost.

All qualitative data (interview transcripts and open-text responses) will be uploaded into NVivo V.12 (QSR International). Interview data will be analysed first; standard care participant interviews will be analysed to gain an understanding of the experiences of participants who receive standard advice for weight loss maintenance on completion of a behavioural weight management programme. Interviews conducted at mid-intervention (3 months from baseline) and postintervention (6 months from baseline) will be compared within and then between standard care participants to understand how their experiences and needs may have changed over the course of the study period. This process will be repeated for the intervention participant interviews, followed by comparison of findings with the standard care participant interviews for convergence and dissonance in experiences and

opinions (eg, standard care participant interviews at 3 months from baseline will be compared with intervention participant interviews at 3 months from baseline, and this process will be repeated for interviews at 6 months from baseline). This will allow us to draw comparisons of experiences and identify how the SWiM intervention may have produced change, compared with standard advice. Next, the interviews with coaches will be analysed and mid- (3 months from baseline) and postintervention (6 months from baseline) interviews will be compared within and between coaches to gain insight into how their experiences and needs may have changed over the course of the study. Coach interviews will then be compared with the participant interviews for convergence and dissonance in views and opinions.

A thematic framework will be generated on the basis of the participant interview data and will be used to guide coding for the open-text responses from the postintervention questionnaires, followed by the open-text responses for the remaining qualitative data from the coach support forms (completed by participants), SWiM session feedback questionnaires (completed by participants) and coach call reports (complete by coaches). While coding the open-text responses, the thematic framework will remain open to the integration of new codes identified throughout analysis.

## Integration of findings

Once the quantitative and qualitative datasets have been analysed separately, the findings will be systematically compared with assess the degree of: (1) agreement (convergence), (2) the extent to which findings offer complementary information on the same issue (complementarity) or (3) appear to contradict one another (dissonance), as well as identify areas of 'silence', where

a theme may be generated from one dataset but not another.[15] This process will strengthen understanding of the findings and themes generated from each method and dataset, and lead to the generation of meta-themes that cut across the datasets.[16] Findings will be summarised and displayed in a table, highlighting the different methods used for each component of the MRC framework.

## PATIENT AND PUBLIC INVOLVEMENT

Substantial patient and public involvement (PPI) has been sought in the development of the SWiM intervention. The initial ideas and research proposal were reviewed by 22 men and women attending the Fakenham Weight Management Service in Norfolk, England and six members of the University of Cambridge PPI Panel. Once funding was awarded, a Patient User Group Panel (PUGP) was formed, comprising members with diverse experiences of weight loss and weight maintenance. This group helped in development and refinement of the logic model, the intervention content and prototype iterations. Regular meetings were convened during the development phase in order to consult on progress and to feedback to the panel on how their input was incorporated. A series of 'think aloud' user testing sessions were completed for the alpha version of the web platform with feedback requested on content, design and functionality. To get feedback on the content from a wider and more diverse audience, remote user testing of the web platform was conducted with the PUGP members and participants from the WRAP study[4] from around the UK.

To maximise participant engagement and retention, and minimise burden, PPI representatives reviewed the content, design and delivery of participant-facing materials. They will also advise on the content and methods of qualitative interviews and focus groups to ensure sensitivity and to maximise participant engagement.

A member of the PUGP will review a sample of the interview transcripts for each participant group and coaches, providing input to the analysis and interpretation of the findings. They will be included as a co-author on the qualitative results paper.

A PPI representative is a member of our Investigator team and has contributed to the design of the protocol and chairs the PUGP. She will contribute to designing and delivering PPI training, preparing ethics and Research and Development submissions, coauthoring journal articles and the final report, disseminating findings to a wide range of audiences and supporting other PPI members.

Two PPI representatives are members of the Programme Steering Committee (PSC). They will review the final study reports and contribute to the writing of specific sections, such as the lay summary.

Including PPI perspectives in plans for dissemination will ensure that we access an appropriate range of audiences and communicate messages effectively. PPI representatives will advise on content and methods of dissemination and will review public-facing documents, such as newsletters and press releases.

PPI representatives will be reimbursed for their time and expenses in a timely manner and tailored PPI training will be provided to suit the specific needs of the individual and their role, guided by INVOLVE Standards for PPI.[17]

## PROGRAMME STEERING COMMITTEE

The PSC will provide overall supervision for the SWiM feasibility study on behalf of the trial sponsors (NHS Cambridgeshire and Peterborough CCG, University of Cambridge) and trial funder (NIHR Clinical Commissioning Facility) and ensure that the project is conducted to the rigorous standards set out in the UK Policy Framework for Health and Social Care Research and the Guidelines for Good Clinical Practice.[18] The PSC will provide advice to the investigators on all aspects of the trial and will review and agree the trial protocol, the statistical analysis plan and any amendments to the protocol. The PSC will be chaired by Professor Lucy Yardley (University of Southampton), who supersedes Professor Andrew Farmer (University of Oxford). Independent members include Dr Thomas Fanshawe (University of Oxford), Dr Edel Doherty (NUI Galway), Mr Graham Rhodes (PPI representative) and Mrs Hazel Patel (PPI representative) who supersedes Mrs Norma Scullion. This is a low-risk trial with no rules for early stopping and participants and study coordinators are not blind to intervention allocation. Thus, a separate data monitoring committee was not deemed to be necessary.

## ETHICS AND DISSEMINATION

Ethical approval was received from Cambridge South Research Ethics Committee on 15 March 2021 (21/EE/0024). At the end of the trial, we will present our findings to our PPI representatives and our stakeholder panel, and we will work with them to refine the intervention and plans for a full-scale trial and to identify appropriate ways to communicate findings to participants and other non-academic audiences.

All specified analyses will be written up as scientific papers and submitted for publication in peer-reviewed open-access journals. Members of the research team will be involved in reviewing drafts of the manuscripts, abstracts and any other publications arising from the trial. The principal investigators will have final approval on all publications and press releases. Authorship will be determined using International Committee of Medical Journal Editors criteria.

## TRIAL STATUS

This protocol (V.2.0) was approved on 19 April 2021. Recruitment for the trial began in May 2021.

**Author affiliations**
[1]MRC Epidemiology Unit, School of Clinical Medicine, University of Cambridge, Cambridge, UK
[2]Fakenham Medical Practice, Fakenham, UK
[3]Faculty of Medicine and Health Sciences, University of East Anglia, Norwich, UK
[4]School of Medicine, University of Leeds, Leeds, UK
[5]Department of Public Health and Primary Care, University of Cambridge, Cambridge, UK
[6]School of Health and Related Research, The University of Sheffield, Sheffield, UK
[7]Quality and Outcomes of Person-Centred Care Policy Research Unit, Canterbury, UK

**Acknowledgements** We thank all staff from the MRC Epidemiology Unit Function Group Team for input into the protocol design, particularly with regard to study coordination, anthropometry measurement, data management, IT, business operations and research governance. We also thank Cambridgeshire and Peterborough CCG Research Team for their input into the operational aspects of this study design and their comments on early iterations of the protocol.

**Contributors** ALA and SG are joint principal investigators. ALA, SG, SS, RD, SM, AH, CH, AB and JB are grant holders. JW is the trial coordinator. ALA, JW, RR, RAJ, JM, FW, SS, RD, AH, SM, CH, AB, JB and SG contributed to the design of the study protocol. ALA and JW wrote the first draft of the manuscript. ALA, JW, RR, RAJ, JM, FW, SS, RD, AH, SM, CH, AB, JB and SG contributed to the writing and critical revision of the manuscript.

**Funding** This study was funded by the National Institute for Health Research (NIHR) under its Programme Grants for Applied Research Programme (RP-PG-0216-20010).

**Disclaimer** The views expressed are those of the author(s) and not necessarily those of the NIHR or the Department of Health and Social Care.

**Competing interests** ALA is principal investigator on another publicly funded trial where JW provided the intervention at no cost but has received no personal fees. SG reports grants from Medical Research Council, personal fees from Eli Lilly and personal fees from Janssen, outside this programme of research. AH reports personal fees from Slimming World and the College of Contemporary Health, outside this programme of research. CH reports informal unpaid advice to Thriva.

**Patient and public involvement** Patients and/or the public were involved in the design, or conduct, or reporting, or dissemination plans of this research. Refer to the Methods section for further details.

**Patient consent for publication** Not applicable.

**Provenance and peer review** Not commissioned; externally peer reviewed.

**Data availability statement** This is a protocol and no new data has yet been created for this study

**ORCID iDs**
Amy L Ahern http://orcid.org/0000-0001-5069-4758
Rebecca Richards http://orcid.org/0000-0001-7122-6822
Julia Mueller http://orcid.org/0000-0002-4939-7112
Robbie Duschinsky http://orcid.org/0000-0003-2023-5328
Emma Ruth Lawlor http://orcid.org/0000-0002-0742-0476
Stephen Morris http://orcid.org/0000-0002-5828-3563

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
