## [Reviewer comments · BMJ Open]

ARTICLE DETAILS

TITLE (PROVISIONAL)	Acceptability and feasibility of an acceptance and commitment therapy-based guided self-help intervention for weight loss maintenance in adults who have previously completed a behavioural weight loss programme: the SWiM feasibility study protocol.
AUTHORS	Ahern, Amy L.; Richards, Rebecca; Jones, Rebecca; Whittle, Fiona; Mueller, Julia; Woolston, Jenny; Sharp, Stephen; Hughes, Carly; Hill, Andrew; Duschinsky, Robbie; Lawlor, Emma; Morris, Stephen; Fusco, Francesco; Brennan, Alan; Bostock, Jennifer; Griffin, Simon

VERSION 1 – REVIEW

REVIEWER	Lillis, Jason Alpert Medical School of Brown University, Psychiatry and Human Behavior
REVIEW RETURNED	22-Nov-2021

GENERAL COMMENTS	This is a well thought out study and a very well-written manuscript. By and large, all the necessary details were provided. Overall, an excellent job. There are a few very minor concerns listed below. Introduction I'm not sure it is accurate to say "interventions based on acceptance and commitment therapy (ACT) may be effective for long-term weight control." It would be more accurate to say that "interventions that incorporate strategies from acceptance and commitment therapy (ACT) may be effective..." As far as I am aware, there is no purely ACT intervention for weight control. There are many tests of behavioral weight loss (BWL) interventions that include/incorporate/integrate ACT methods that show efficacy for weight loss and promise for long-term weight loss. Inclusion/Exclusion criteria No mention of a BMI maximum or minimum. If there is none, perhaps say why that is. Will you accept a BMI 17 person who recently completed a commercial programme? No mention of verification of programme participation, or criteria for having lost weight in a programme (even if just self-report). If there are no criteria for those, perhaps explain why. If the SWiM is focused on maintenance, will you accept a person who recently enrolled in weight loss for 3 months, but gained a significant amount of weight? Justification for over-inclusiveness would be helpful here.
--

	Recruitment Minimal details provided. Will the researchers be partnering with specific programmes? How will the study be presented to potential participants? By study staff or by weight loss groups leaders? SWiM Content There is essentially only an outline. No description of the content, no examples. Perhaps that is OK for this protocol paper, but often protocol papers are used to be more descriptive about the intervention than you can be in an empirical report after study completion. Coaches and phone calls Minimal details provided. Who will monitor training coaches, or will it be fully automated? Is the training procedure/ materials validated or is this something being created for the current study? Details about the training procedures would be helpful.
--	--

REVIEWER	Kos, K University of Exeter, Diabetes and Obesity Research
REVIEW RETURNED	29-Dec-2021

GENERAL COMMENTS	The novelty of this study is to appreciate the importance of weight maintenance and follow up after completion of standard weight management. The protocol is explained elaboratively and I have a few minor questions including a request for some clarifications on items which may be important to consider for the feasibility trial and/or later trial refinements. Recruitment of subjects: completion of weight loss programme is an inclusion criterion, does this necessitate 'sufficient' engagement/attendance or success in regards of weight loss? Given that some patient stabilise their weight and some gain weight in that period, do they equally qualify? How about those who continue with commercial weight loss groups and VLCDs? Please clarify also whether subjects who use GLP-1 analogues (before or during trial) which suppresses appetite should be included and the rational for excluding those on insulin therapy. Whilst depression and anxiety questionnaires are being used, there is no exclusion of individuals with mental health issues and/or learning difficulties. Some eating disorders become apparent only during weight loss interventions and require explicit screening rather than relying on history e.g. from the previous weight management team. Will subjects with past eating disorders and/or recent therapy qualify? Intervention: How much time is the participant expected to spend on an individual session, e.g. as in participant information? Coaches will be necessary to enable motivation to adherence and completion of the intervention. Should this become a clinical intervention their integration in a service I expect will be crucial. How will these be explicitly accounted to the service cost?
---

VERSION 1 – AUTHOR RESPONSE

Reviewer: 1

Dr. Jason Lillis, Alpert Medical School of Brown University.

Comments to the Author:

This is a well thought out study and a very well-written manuscript. By and large, all the necessary details were provided. Overall, an excellent job. There are a few very minor concerns listed below.

RESPONSE: We thank the reviewer for their careful consideration of this paper and their constructive appraisal.

Introduction

COMMENT: I'm not sure it is accurate to say "interventions based on acceptance and commitment therapy (ACT) may be effective for long-term weight control." It would be more accurate to say that "interventions that incorporate strategies from acceptance and commitment therapy (ACT) may be effective..." As far as I am aware, there is no purely ACT intervention for weight control. There are many tests of behavioral weight loss (BWL) interventions that include/incorporate/integrate ACT methods that show efficacy for weight loss and promise for long-term weight loss.

RESPONSE: We understand the reviewer's thoughts on this. We use the term ACT-based to describe interventions where ACT is the dominant feature of the intervention, rather than a pure ACT intervention. Having evolved from traditional CBT, many ACT-based interventions also include more traditional behavioural elements and in the current intervention we have incorporated some of these in a way that is in keeping with the ACT ethos. However, we agree that when referring to the work of others, who perhaps started with the traditional behavioural programme as the starting point and then added ACT elements, the phrasing "interventions that incorporate strategies from acceptance and commitment therapy (ACT)" is more precise. We have made this change on page 2.

Inclusion/Exclusion criteria

COMMENT: No mention of a BMI maximum or minimum. If there is none, perhaps say why that is. Will you accept a BMI 17 person who recently completed a commercial programme?

RESPONSE: We apologise for this omission. There is no maximum BMI for participants in order to be as inclusive as possible. Typically, participants are eligible for behavioural weight loss programmes if they have a BMI of 25 kg/m² or above (or 23 kg/m² depending on ethnicity). It is therefore unlikely that participants would lose so much weight on a behavioural weight loss programme to reach a BMI where they are considered 'underweight' (i.e. BMI of <18.5 kg/m²). Should this situation occur, we would identify this at screening and we would then make a judgement on whether our SWiM maintenance programme is suitable for this participant. One of the aims of this feasibility study is to determine the population for which the SWiM programme is suitable, and so further exclusion and inclusion criteria will be developed prior to a full trial. We have added the following text to the section titled "Exclusion criteria", on page 5:

"There are no inclusion or exclusion criteria for BMI in order to be as inclusive as possible. Participants typically require a BMI of 25 kg/m² or higher (or 23 kg/m² depending on ethnicity) to be eligible for behavioural weight loss programmes, therefore we do not expect that participants will lose enough weight to reduce their BMI to 18.5 kg/m² or less (i.e. 'underweight') and participate in the SWiM programme. Any potential participants who are underweight would be identified at screening

and further assessment conducted to determine suitability. This feasibility study will inform whether further inclusion and exclusion criteria should be implemented prior to a full trial.”

COMMENT: No mention of verification of programme participation, or criteria for having lost weight in a programme (even if just self-report). If there are no criteria for those, perhaps explain why. If the SWiM is focused on maintenance, will you accept a person who recently enrolled in weight loss for 3 months, but gained a significant amount of weight? Justification for over-inclusiveness would be helpful here.

RESPONSE: We have not included a requirement for verification of participation or weight loss in a behavioural weight loss programme because these would represent an additional burden on participants or referring programmes that might hinder efficient referral and limit intervention uptake. Even if no weight was lost in the original weight loss programme, the skills and strategies learned during SWiM could support participants’ learnings from their original programme and could help them to lose weight or prevent further weight gain. Participants will self-report any weight change during their original programme as part of the first SWiM session. We will be able to consider the frequency and impact of this issue further as part of the process evaluation component of this feasibility study. We have added the following text to the section titled “Inclusion criteria”, on page 4:

“There is no requirement for verification of participation in a behavioural weight loss programme in the last 3 months in order to avoid unnecessary additional burden on participants or referring programmes and to support efficient referral and wide uptake. There are no criteria for having lost weight during a behavioural weight loss programme as the SWiM programme can be used to support and build upon participants’ learnings from their behavioural weight loss programme and participants can develop further skills and strategies from ACT that could help them to lose weight or prevent further weight gain.

Recruitment

COMMENT: Minimal details provided. Will the researchers be partnering with specific programmes? How will the study be presented to potential participants? By study staff or by weight loss groups leaders?

RESPONSE: As this is a feasibility study, we did not limit recruitment to a specific programme. We have added more details to the “Recruitment” section on page 5 to say:

“Participants will be recruited through NHS, local authority and commercial weight management services and diabetes prevention programmes. Eligible people will be given or mailed a study information leaflet by their programme leader within their weight or type 2 diabetes management programme”.

SWiM Content

COMMENT: There is essentially only an outline. No description of the content, no examples. Perhaps that is OK for this protocol paper, but often protocol papers are used to be more descriptive about the intervention than you can be in an empirical report after study completion.

RESPONSE: We have recently published a manuscript containing full details of the intervention development process and content, including images from the intervention and screenshots of content (<https://formative.jmir.org/2022/1/e31801>). We have now added the following text to the section titled “Planned intervention and standard care” on page 6: “Full details of intervention content have been published elsewhere [8]”. Readers can refer to this manuscript for further details.

Coaches and phone calls

COMMENT: Minimal details provided. Who will monitor training coaches, or will it be fully automated?

Is the training procedure/ materials validated or is this something being created for the current study? Details about the training procedures would be helpful.

RESPONSE: The following text has been added to the section titled "Planned intervention and standard care", on page 6: "Coaches are trained by a member of the research team, who is a Practitioner Psychologist (RR), and the training was developed by the research team specifically for this study. As part of the training, coaches are asked to: i) read through a training manual, which includes semi-structured scripts to guide telephone calls, and ii) attend three hours of training with the lead Practitioner Psychologist either in person or via video call. The training manual includes an outline of intervention content, underpinning theories of the SWiM programme (ACT and motivational interviewing), a practical guide to conducting telephone support, the procedures for participant withdrawal, information on how to avoid stigmatising language and information for signposting participants to mental health support. The Practitioner Psychologist training takes coaches through the training manual in detail, provides opportunities for questions and includes role-play of each telephone call script with feedback. Following completion of this training, coaches have the opportunity for one-to-one or group follow up sessions for further practice or clarifications. Coaches are also asked to complete each session on the SWiM website from the perspective of the participant."

Reviewer: 2

Dr. K Kos, University of Exeter

Comments to the Author:

COMMENT: The novelty of this study is to appreciate the importance of weight maintenance and follow up after completion of standard weight management. The protocol is explained elaboratively and I have a few minor questions including a request for some clarifications on items which may be important to consider for the feasibility trial and/or later trial refinements.

RESPONSE: We thank the reviewer for their careful consideration of this paper and their constructive appraisal.

COMMENT: Recruitment of subjects: Completion of weight loss programme is an inclusion criterion, does this necessitate 'sufficient' engagement/attendance or success in regards of weight loss? Given that some patient stabilise their weight and some gain weight in that period, do they equally qualify?

RESPONSE: As described in our response to Reviewer 1 above, we have made changes to the manuscript to clarify that there are no criteria for having lost weight during a behavioural weight loss programme as the SWiM programme can be used to support and build upon participants' learnings from their original behavioural weight loss programme and develop further skills and strategies from ACT that could help them to lose weight or prevent further weight gain, depending on their individual goals.

COMMENT: How about those who continue with commercial weight loss groups and VLCDs? Please clarify also whether subjects who use GLP-1 analogues (before or during trial) which suppresses appetite should be included and the rationale for excluding those on insulin therapy.

RESPONSE: We expect that some participants may continue to receive support in the real world (e.g. attending groups, using other support such as apps to monitor food intake) and we do not wish to exclude these participants. We will collect information on whether participants continue to receive any support from their original behavioural weight loss programme or additional interventions. If participants state that they are continuing with a more intense intervention, such as VLCDs, or start a medical intervention, such as GLP-1 analogues, during the feasibility study, we will judge their

eligibility to participate/continue on a case-by-case basis. One of the aims of this feasibility study is to determine the population for which the SWiM programme is most suitable. We have added the following text to the section “Exclusion criteria”, on page 5:

“We expect that some participants may continue to receive weight management support whilst participating in the SWiM programme (e.g. attending groups, using an app to monitor food intake) and we do not wish to exclude these participants. We will collect information on whether participants continue to receive any support from their original behavioural weight loss programme or additional interventions. If participants report that they are receiving a more intense intervention, such as very low calorie diets, or a medical intervention, such as GLP-1 analogues, during the feasibility study, we will judge their eligibility to participate and/or continue on a case-by-case basis. This feasibility study will inform whether further inclusion and exclusion criteria should be implemented prior to a full trial.”

Our expert stakeholder panel advised that the specific support needs of adults using insulin were beyond the remit of this intervention. We have updated the “Exclusion criteria” section (page 4) to include this information:

“Based on expert stakeholders advice that the specific support needs of the following groups are beyond the remit of this intervention, adults who meet any of the following criteria will not be eligible for inclusion: i) Using insulin, ii) Previous or planned bariatric surgery, iii) Current or planned pregnancy, iv) Current diagnosis of eating disorder.”

COMMENT: Whilst depression and anxiety questionnaires are being used, there is no exclusion of individuals with mental health issues and/or learning difficulties. Some eating disorders become apparent only during weight loss interventions and require explicit screening rather than relying on history e.g. from the previous weight management team. Will subjects with past eating disorders and/or recent therapy qualify?

RESPONSE: We are recruiting people who have already taken part in a behavioural weight management programme, therefore we do not feel that it is appropriate to add further exclusion criteria for participation in SWiM for things such as mental health or learning difficulties. We expect that those who have successfully completed a standard behavioural programme will be able to participate in SWiM. However, this is a feasibility study and we will be able to assess the suitability of this approach. It is not standard practice to conduct repeated assessment of eating disorders during weight management treatment and so we would only screen people for eating disorder diagnosis at baseline. However, we would expect participants to report any difficulties that they may experience during the study to their coach or our research team. We would then make a judgement on whether to withdraw the participant and signpost them to the appropriate support. If participants disclose a recent diagnosis of an eating disorder to their coach or our research team, we would withdraw the participant from the study and this would be formally recorded as a medical event occurring during the trial. There is surprisingly little research on the relationship between weight management treatment and eating disorder outcomes, with many studies showing that on average behavioural interventions reduce eating disorder symptoms. Ahern and Hill are part of the Eating Disorders in Weight Related Therapy (EDIT) collaboration (<https://www.editcollaboration.com/>) which seeks to identify the intervention components and individual characteristics that might increase risk of eating disorders and we will ensure that any findings from this work are applied to future work on the SWiM programme.

COMMENT: Intervention: How much time is the participant expected to spend on an individual session, e.g. as in participant information?

RESPONSE: We have added the following text to the section titled “Planned intervention and standard care”, on page 6: “Each session is expected to take around 30-60 minutes to complete”.

COMMENT: Coaches will be necessary to enable motivation to adherence and completion of the intervention. Should this become a clinical intervention their integration in a service I expect will be crucial. How will these be explicitly accounted to the service cost?

RESPONSE: Thank you for your question. Resource use for the set-up, delivery and maintenance of the programme (including the integration of SWiM coaches) will be estimated via a micro-costing. We have already started this process, with resource use estimates derived from the literature, but this will be refined following the feasibility study and adjusted based on expert input through an informal panel. We recognise that these details were not included in the study protocol and have added these to the manuscript on page 11, under the section titled "Economic analysis":

"We will conduct a micro-costing to estimate resource use for the set-up, delivery and maintenance of the SWiM programme. Resource use estimates will be derived from the literature and informed by feasibility study data, and will be adjusted based on expert input through an informal panel process."

VERSION 2 – REVIEW

REVIEWER	Kos, K University of Exeter, Diabetes and Obesity Research
REVIEW RETURNED	14-Mar-2022
GENERAL COMMENTS	I reviewed to author responses to my comments and have no further concerns for publications of this feasibility study protocol.